# Assessing the Operational Capability of Disaster and Emergency Management Resources: Using Analytic Hierarchy Process

**Ke Zhang** [ID] **and Jae Eun Lee \***[ID]

Department of Crisisonomy & National Crisisonomy Institute, Chungbuk National University, Chungbuk 28644, Republic of Korea; zhangke35307@naver.com
\* Correspondence: jeunlee@chungbuk.ac.kr; Tel.: +82-43-261-2197

**Abstract:** This study aims to assess the operational capability of disaster and emergency management resources (DEMRs), which is not only critical for effective loss reduction and resilience, but also facilitates the sharing and optimal use of resources for the more effective achievement of sustainable development. This study constructs an evaluation index system of the operational capability of DEMRs, encompassing four key aspects: resource planning, organizational management capability, resource support capability, and information processing capability. It focuses on identifying the factors that influence the operational capability of DEMRs in China and Korea, comparing and analyzing the relative importance and priority of each evaluation domain and indicator within these countries. The results show that the organizational management capability is most significant in China, whereas the resource support capability is prioritized in Korea. A comparative analysis of the local weight of indicators within each domain revealed the largest discrepancy between China and Korea in the information processing capability domain. This study concludes by calculating global weights, identifying the fast response capability and resource allocation capability as the most impactful factors on the operational capability of DEMRs, and highlighting their critical role in disaster and emergency management.

**Keywords:** disaster and emergency management resources; operational capability; AHP; evaluation index system





## 1. Introduction

Disasters are growing more complex and unpredictable, fueled by escalating urbanization and global climate change [1]. Every country worldwide faces unavoidable threats from both man-made and natural disasters, posing severe risks to people's lives, property, and socio-economic progress. One of the core duties of national governments has always been to safeguard the lives and property of their citizens from disaster impacts [2]. Given these multifaceted threats, it is crucial for governments to develop and implement a practical disaster and emergency management system.

The operation of disaster and emergency management resources (DEMRs) plays a crucial role in disaster and emergency management, directly impacting the effectiveness of disaster and emergency relief efforts. This involves the identification, acquisition, allocation, and distribution of resources to address the needs prompted by emergency and disaster situations [3]. The complexity of effectively managing these operations is heightened by the unpredictable nature, widespread devastation, and dynamic evolution of disasters [4–6]. This process is particularly susceptible to a myriad of uncontrollable factors, such as organizational challenges, human elements, material logistics, information dissemination, and environmental conditions, all of which can significantly hinder the progress of emergency rescue operations. To enhance the coordination of disaster relief resources and ensure the needs of emergency and disaster relief are met for government departments, identifying the key factors influencing the effective operation of DEMRs is essential.

Research in disaster management, emergency response, and emergency resources has gained significant attention from both government bodies and academic institutions. In terms of government practices, the United States pioneered the evaluation of emergency response capabilities and established the National Incident Management System (NIMS) in 2004, which includes guidelines for emergency resource management. Japan has long focused on disaster response and recovery efforts, implementing an evaluation system for disaster prevention capabilities, including a resource management system, in 2002 [7,8]. South Korea (hereinafter Korea) has advanced in evaluating the emergency management capacity, with annual assessments conducted by the Ministry of Administration and Security that cover disaster management tasks and the operation of disaster management resources [9]. In China, the emphasis has been on the issuance of relevant policies and normative documents. The National People's Congress of China approved the 14th Five-Year Plan in March 2021, highlighting the need to strengthen the emergency supplies guarantee evaluation system.

Numerous scholars have highlighted the pivotal role disaster and emergency management resources (DEMRs) play in effectively managing crises. Zhai and Lee [10] have underscored the critical importance of disaster management resources in ensuring preparedness. Similarly, Miao et al. [11] pointed out the essential role of emergency resource management in mitigating losses from natural disasters, while Kim et al. [12] have argued for the necessity of early preventive actions through disaster management resources to limit the extent of disaster impacts. Additionally, there is a significant focus on enhancing emergency rescue efforts through analyses of the storage, logistics, and distribution of emergency resources. For instance, Feizollahi et al. [13] conducted an empirical study to identify crucial factors in emergency logistics, utilizing the Analytic Hierarchy Process (AHP) to rank important activities for optimizing logistic operations. Ma et al. [14] explored how intelligent technologies influence emergency resource allocation using the entropy–TOPSIS method.

Despite the emphasis on the significance of DEMRs in disaster and emergency management, there is a noted gap in research on their operational capability, often with a narrow focus on singular indicators. The innovation of this paper is the incorporation of resource planning and organizational management capacity into the evaluation indicators. Furthermore, in the context of the Fourth Industrial Revolution, the support of information technology in relief efforts, alongside advanced technical monitoring and warning systems, has been identified as a crucial success factor [15]. Therefore, the evaluation of the operational capabilities of DEMRs should also encompass information processing capabilities, addressing the need for a systematic and comprehensive approach to capability building. Another innovation of this paper is the exploration of the differences in these factors between China and Korea—a gap not yet explored in the literature. Both China and Korea are situated in East Asia and share similar geographical and climatic characteristics. Due to their proximity, they also face common challenges and vulnerabilities related to disasters such as earthquakes, floods, typhoons, and infectious diseases. Both China and Korea have implemented various disaster risk reduction and response policies and measures. Analyzing the differences in their perceived priorities can enable policymakers and practitioners to understand areas that may need improvement or adjustment to enhance disaster resilience and response capabilities.

This study seeks to ascertain the relative importance and priority of factors that influence the operational capabilities of DEMRs, and to explore how these factors vary between China and Korea. To achieve these goals, this research begins with a thorough review of the literature to pinpoint specific factors impacting the effectiveness of DEMRs across four key dimensions: resource planning, organizational management capability, resource support capability, and information processing capabilities. Next, employing a combination of expert surveys and the Analytic Hierarchy Process (AHP), this study refines and develops an index model to assess the relative importance and priority of various domains and indicators related to the operational capabilities of DEMRs. This paper concludes by highlighting major findings and offering recommendations to bolster

the effective operational capacity of DEMRs. It aims to provide theoretical insights for improving the operational efficiency and collaborative efforts of DEMRs in China and Korea, thereby offering ongoing support for the advancement of the disaster and emergency management capability in both countries.

## 2. Literature Review

### 2.1. Disaster and Emergency Management Resources (DEMRs)

In 2023, the Korean government enacted the "Disaster Management Resources Management Act" to protect the lives and property of its citizens. This act categorizes disaster management resources as the essential materials, properties, and human resources required for efficient disaster management. Meanwhile, according to China's "National Overall Emergency Response Plan for Emergencies," emergency resources encompass a broad range of assets, including human, material, financial, facilities, information, technology, and other resources necessary to ensure the effective execution of emergency activities and the smooth functioning of the emergency management system.

Numerous scholars have delved into the realm of disaster and emergency resources. In Korea, Kim et al. [12] have categorized disaster resources into human, equipment, and material categories, highlighting issues such as resource scarcity, capacity, utilization, and response time in disaster management. Lee et al. [16] consider disaster prevention resources to encompass human resources, materials, equipment, and facilities mobilized during disasters. In China, Zhou and She [17] describe emergency resources as a broad spectrum of essential supplies, relief equipment, and basic necessities for emergency rescue operations. Qin et al. [18] differentiate emergency resources into response and recovery categories, based on their use in different stages of emergency management. Shao et al. [19] emphasize that emergency resources form the foundational support for disaster and emergency management, playing a crucial role in the success of emergency responses.

This paper notes the variability in terminology used by governments and scholars in both countries regarding resources critical for disaster or emergency response and system functionality. These resources, which include human, material, financial, facilities, information, and technology, are collectively referred to as DEMRs. DEMRs represent a comprehensive term encompassing various resources that can be quickly mobilized or positively responded to in a short time frame during a disaster or emergency. Effective disaster and emergency management necessitates the integration of diverse societal resources, coordinating all necessary activities to mitigate hazards timely and efficiently [20].

### 2.2. The Operation of DEMRs

Emergency response involves not just providing ample resources but also their effective management, which is crucial for making emergency responses more orderly and improving the overall effectiveness of interventions [21]. Establishing a comprehensive resource management process is key to aligning resource capabilities, enhancing coordination, and ensuring interoperability nationwide. According to the National Incident Management System (NIMS), emergency resource management involves the application of processes, personnel, and tools to orchestrate the use of resources such as personnel, teams, facilities, and equipment. Its primary goal is to help policymakers optimize the use of emergency management resources to minimize damage and save lives [22]. Kim et al. [23] have categorized emergency resource management into three main types: equipment, supplies, and human resources, further breaking them down into 11 collaborative functions including life support, energy support, facility emergency recovery, and emergency communication support. Miao et al. [11] highlight the importance of emergency resource management in disaster response, viewing it as a crucial aspect of building resilience. Rodríguez et al. [24] point out that the successful logistical deployment of resources to aid disaster victims heavily depends on the collaboration among various organizations and participants.

In this study, based on earlier research, we defined the function of DEMRs as the coordination among various government departments and social organizations. This coor-

dination aims to ensure the rapid and accurate distribution of essential emergency supplies from areas of availability to areas of need, minimizing the time taken. DEMRs encompass a range of activities designed to enhance the effectiveness of emergency responses and mitigate the negative impacts of disasters. Furthermore, we contend that a comprehensive perspective is crucial when considering the factors and requirements of the operations of DEMRs. Effective coordination between government entities, individuals, and organizations is essential for the swift mobilization of resources. Additionally, incorporating information technology innovations is vital for enhancing the operational efficiency of these resources.

### 2.3. Construction of the Evaluation Index System of the Operational Capability of DEMRs

The evaluation index system is a set of two or more indexes used to effectively evaluate the performance, effectiveness, or capacity of a specific system [25]. For disasters or emergencies, it is critical to identify changing resource needs in the disaster response environment and develop the operational capabilities of the DEMRs necessary to respond to the disasters and emergencies [26]. Improving the operational capability of DEMRs has become an important research topic. Cigler [27] defines capabilities as the assortment of financial, technological, policy, institutional, leadership, and human resources that government agencies must have to effectively manage all phases of emergency response. Kusumasari et al. [26] view resource management capability as a combination of institutional resources, human resources, policies for effective execution, and financial and technical resources, underpinned by leadership. Consequently, assessing the operational capabilities of DEMRs involves a multi-level, multi-indicator approach that incorporates various factors for a thorough analysis.

Emergency resource planning is critical to managing crisis [28]. Aziz et al. [29] identified the prioritization of resilience criteria and performance indicators for road emergency crisis response, and response planning was the highest ranked criterion overall, with joint response planning and resource planning being equally prioritized sub-criteria. Zhai and Lee [10] constructed evaluation indexes for the disaster preparedness capacity of local governments, with three sub-criteria of disaster risk assessment, disaster response planning, and the preparation and approval of planning included in the planning.

Organizational management capability involves managers arranging goals, tasks, and decisions effectively. They create suitable structures and teams, blend resources efficiently, and ensure the smooth execution of decisions [30]. Zhang et al. [30] designed an evaluation index system for emergency logistics capacity, in which the emergency organization and management capability includes five indicators: scientific decision level, overall coordination ability, command and dispatch capability, fast response capability, and social mobilization capacity. Wang and Zhang [31] presented an assessment model focused on the supply of relief supplies in disaster areas. The model assesses the humanitarian relief goods supply capability and emphasizes the important role of decision-making agents and execution agents in regulating the supply of relief supplies as well as rapid coordination.

In terms of resource support capability, Huang and Shi [32] constructed the evaluation indexes of the food emergency logistics supply capacity under natural disasters, and the emergency food response capacity was designed with several sub-criteria, such as collection capacity, reserved capacity, transportation capacity, distribution capacity, delivery timeliness, and rationality of the logistics center setup. Wang et al. [33] constructed evaluation indexes for the emergency management capacity of disaster-resistant communities, in which emergency material support contains secondary indexes such as emergency material saving points, family storage, and social storage.

In terms of information processing capability, information technology is an important symbol of disaster and emergency management, which plays an important role. Xu and Gong [34] established an evaluation index system for emergency logistic support capability consisting of three parts: command and control capability, material management capability, and information management capability. The evaluation indexes of information manage-

ment capability were proposed to have an emergency monitoring and prediction ability, information collection and analysis ability, and comprehensive database. Du et al. [35] developed a multilevel indicator system to measure equipment assurance capability. This includes transportation information capability evaluation indicators such as information acquisition capability, processing capability, and transmission capability.

The principles of objectivity, systematicity, comprehensiveness, and coordination in constructing the indicator system should be followed to effectively improve the credibility of the evaluation results [36]. Based on national policies and literature review, this paper starts with the concept of DEMRs. It draws on NIMS resource management guidance, the "Disaster Resource Management Act" in Korea, and the "Emergency Resource Assurance Plan" in China to construct an evaluation index system of the operational capability of DEMR with four domains and 16 indicators, as shown in Table 1.

**Table 1.** Description of the evaluation index system of the operational capability of DEMRs.

| Domain | Indicator | Description | References |
|---|---|---|---|
| Resource planning | Policy guidance | Formulate and implement emergency response policies to provide a framework and principles guiding the resources operation in emergency situations. | [10,29–35] |
| | Demand assessment | Evaluate potential demand in emergency situations and provide data support for plan formulation. | |
| | Resource operation plan | Develop clear resource operation plans to ensure effective allocation, utilization, and monitoring of resources. | |
| | Funding budget | Establish funding budgets to ensure adequate economic support. | |
| Organizational management capability | Command and dispatch capability | Effectively command and dispatch organizations and personnel at all levels, ensuring coordinated response activities. | |
| | Fast response capability | Respond rapidly, flexibly, and efficiently to emergency situations to mitigate losses and expedite post-disaster recovery. | |
| | Social mobilization capability | Effectively mobilize resources and support from all sectors of society, forming a collective effort. | |
| | Communication capability | Timely and accurate information transfer between the organization and stakeholders. | |
| Resources support capability | Reserve capability | Effectively stock and manage various resources (material, equipment, human resources, technology, information) required in emergency situations | |
| | Transportation capability | Establish an efficient resource transport system to ensure timely and safe delivery of resources to the designated locations. | |
| | Scheduling capability | Efficiently schedule various resources to ensure their reasonable allocation at different locations and times. | |
| | Allocation capability | Flexibly and efficiently allocate various resources to meet the actual needs of different regions and departments. | |
| Information processing capability | Early warning technology | Use advanced technological means to detect potential risks and threats early, providing timely and accurate warning information. | |
| | Timely information acquisition | Rapid retrieval and timely transmission of critical information related to resource operations to support decision making and effective operations. | |
| | Information-sharing capability | Governments and stakeholders effectively share critical information about emergencies and resources. | |
| | Monitoring and tracking of logistics | Implement effective logistics monitoring systems to track and manage the transportation, distribution, and use of resources. | |

## 3. Materials and Methods

### 3.1. Analytic Hierarchical Process (AHP)

The AHP is a decision-making methodology that integrates both quantitative and qualitative analysis, introduced by Saaty in the 1970s [37]. This approach assists in extracting preferences from decision makers by breaking down complex, unstructured problems into simpler, multi-level hierarchies. It facilitates pairwise comparisons to determine the relative importance of each element, leading to the ranking of options to guide the selection of the most suitable solution [38]. AHP is particularly useful in scenarios fraught with uncertainty and multiple criteria, finding wide application across government, business, construction, healthcare, and education sectors. In the realm of disaster and emergency management, it aids in areas like disaster preparedness [10,39], emergency response [40,41], risk management and resilience assessment [42–44], sustainability assessment [45], and emergency supply chain risk analysis [46,47].

This research begins with a review of the literature on DEMRs and their operational capabilities. It proposes an evaluation index system for these capabilities, establishing a hierarchical model based on their interrelationships. Researchers were invited to test this model, ensuring the index system's comprehensiveness. Following validation, appropriate experts were chosen for data collection. A judgment matrix was created using the pairwise comparison method, from which the relative importance of each evaluation indicator was calculated. The consistency of these evaluations was verified using the consistency ratio (CR). This study concludes by aggregating the weights of the indicators at different levels to compute the overall priority of each, thereby identifying key factors influencing the operational capability of DEMRs. Finally, a sensitivity analysis was performed to verify the robustness of the results. The detailed methodology is illustrated in Figure 1.

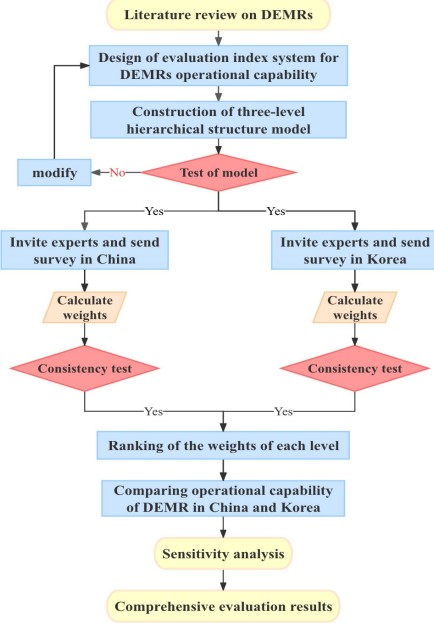

**Figure 1.** The flowchart of evaluation of the operational capability of DEMRs.

### 3.2. Establishment of the Hierarchical Structure Model

The hierarchical structure model for assessing the operational capability of DEMRs consists of three tiers: the target layer (first tier), criteria layer (second tier), and scheme layer (third tier). At the top, the target layer (A) is defined as the operational capability of DEMRs. The criteria layer is segmented into four domains, resource planning (B1), organizational management capability (B2), resource support capability (B3), and information processing capability (B4), each accompanied by four sub-indicators for a comprehensive evaluation. The full evaluation model is depicted in Figure 2.

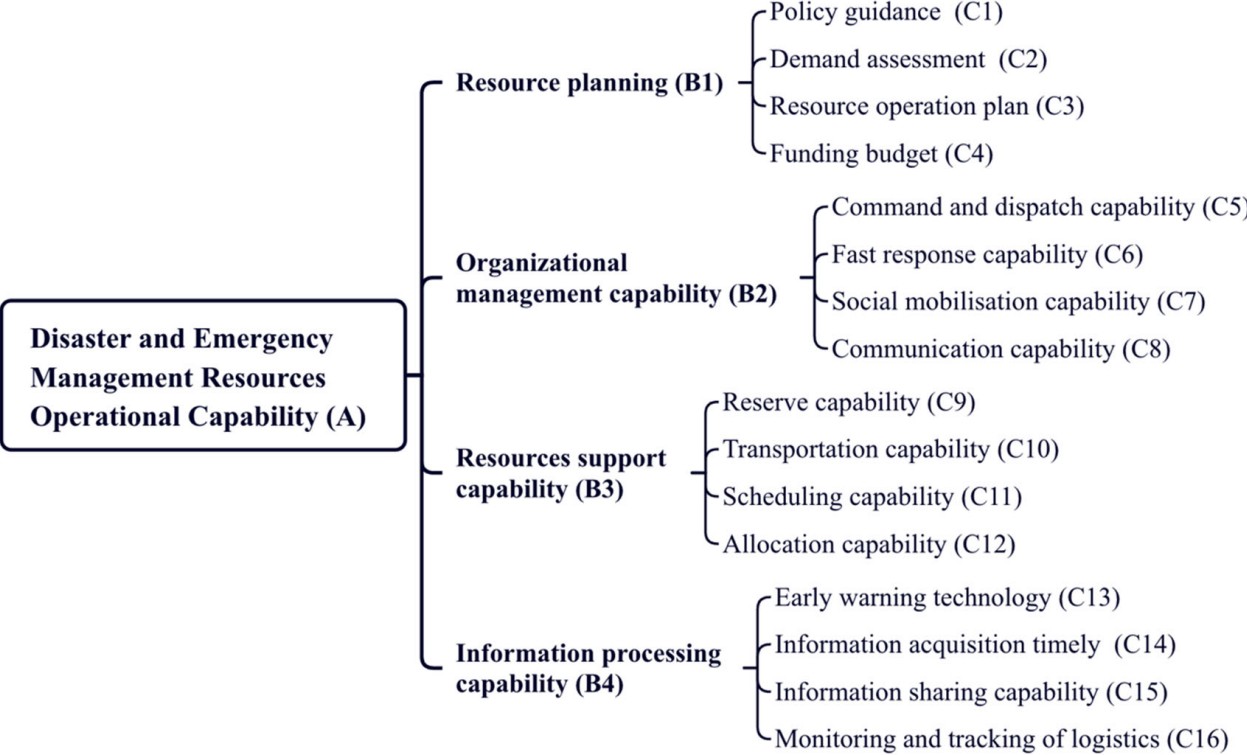

**Figure 2.** The evaluation model of the operational capability of DEMRs.

*3.3. Construction of Judgement Matrix*

Constructing a judgment matrix is a key phase in applying the Analytic Hierarchy Process (AHP) method. To guarantee the matrix's reliability and to scientifically set the importance and priorities of the indicators, the process began by inviting ten Ph.D. researchers to review and adjust the indicators in the first round. The ten Ph.D. researchers are all doctoral candidates or have already obtained doctoral degrees in the fields related to disaster and emergency management, with more than two years of research experience, as shown in Table 2.

**Table 2.** Characteristics of survey participants in the first round.

| Characteristics | | Frequency | Characteristics | | Frequency |
|---|---|---|---|---|---|
| Gender | Male | 6 | Occupation | Ph.D. Student | 7 |
| | Female | 4 | | Assistant Researcher | 3 |
| Age | 20–30 | 5 | Number of years of research | 2–5 years | 6 |
| | 31–40 | 5 | | More than 5 years | 4 |

Then, a group of 22 experts in the field of emergency and disaster management was assembled, including 11 from China and 11 from Korea. They are professors and government staff with extensive experience and a high level of understanding in fields related to disaster and emergency management. Table 3 shows the demographic characteristics of the experts interviewed in the second round. Through an online survey, these experts created pairwise comparison judgment matrices. Their assessment was based on their expertise, knowledge, and practical experience using the 1–9 scale approach suggested by Saaty [37], as shown in Table 4, i.e., assigning a scale value from 1 to 9 to each dimension based on their comparative prominence in this degree [48]. This method allowed the experts to more effectively and quickly evaluate the significance and values of each indicator.

**Table 3.** Characteristics of survey participants in the second round.

|  | Characteristics | From China | From Korea |
|---|---|---|---|
| Gender | Male | 6 | 7 |
|  | Female | 5 | 4 |
| Age | 20–30 | 2 | 1 |
|  | 31–40 | 6 | 8 |
|  | 41–50 | 2 | 1 |
|  | Over 50 | 1 | 1 |
| Occupation | Professor | 4 | 2 |
|  | Government staff | 5 | 4 |
|  | Researchers in scientific institutions | 2 | 5 |
| Number of years of research or work | 5–10 years | 3 | 2 |
|  | 10–20 years | 6 | 5 |
|  | More than 20 years | 2 | 4 |

**Table 4.** Judgment matrix scale.

| Scales | Definition | Interpretation |
|---|---|---|
| 1 | Equal importance | Two indicators have equal importance. |
| 3 | Moderate importance | One is moderately more important than the other. |
| 5 | Strong importance | One is strongly more important than the other. |
| 7 | Very strong importance | One is very strongly more important than the other. |
| 9 | Extreme importance | One is extremely more important than the other. |
| 2,4,6,8 | Intermediate values of two adjacent levels | Adopted for compromising. |

### 3.4. Indicator Weight Calculation and Consistency Test

In this study, we utilized YAAHP software (v12.11) to calculate the weights of the indicators at various levels. To aggregate the personal judgments of experts, the literature suggests employing either the geometric mean or the arithmetic mean [49]. The results of pairwise comparisons were normalized via standard arithmetic operations to form a normalized matrix. After obtaining the normalized values, a consistency test was conducted to ensure compliance with the acceptable consistency conditions [50]. It is widely accepted among researchers that consistency ratios (CRs) of up to 0.10 are considered acceptable. However, some scholars suggest that a limit of up to 0.20 can also be acceptable, but not exceeding that threshold [51,52]. Saaty [37] explicitly defined the calculation of the Consistency Index (CI) for a comparison matrix as $CI = (\lambda max - n)/(n - 1)$. Moreover, he provided specific values for the Random Consistency Index (RI) based on the number of criteria being evaluated [38]. The calculation of the consistency ratio (CR) is then given by the formula $CR = CI/RI$. The application of the AHP model to evaluate the operational capability of DEMRs in China resulted in a CR of 0.072, which is less than 0.1. Similarly, for Korea, the CR was 0.098, also below 0.1. These results, indicating a high level of consistency in the judgment matrices, are detailed in Table 5.

**Table 5.** The results of consistency test.

|  | A | B1 | B2 | B3 | B4 |
|---|---|---|---|---|---|
| China | 0.072 | 0.030 | 0.056 | 0.038 | 0.072 |
| Korea | 0.098 | 0.031 | 0.053 | 0.039 | 0.039 |

## 4. Results

### 4.1. Local Weight Ranking Comparison

Figure 3 presents the weight distribution across various domains in China and Korea. For China, organizational management capability (B2) with a weight of 0.427 was identified as the most critical factor influencing the overall operational capability of DEMRs, followed by resource support capability (B3) with a weight of 0.361, resource planning (B1) at 0.110, and information processing capability (B4) with the lowest weight at 0.102. In Korea, resource support capability (B3) emerged as the most significant factor with a weight of 0.358, followed by organizational management capability (B2) at 0.313, information processing capability (B4) at 0.250, and resource planning (B1) receiving the least emphasis with a weight of 0.079.

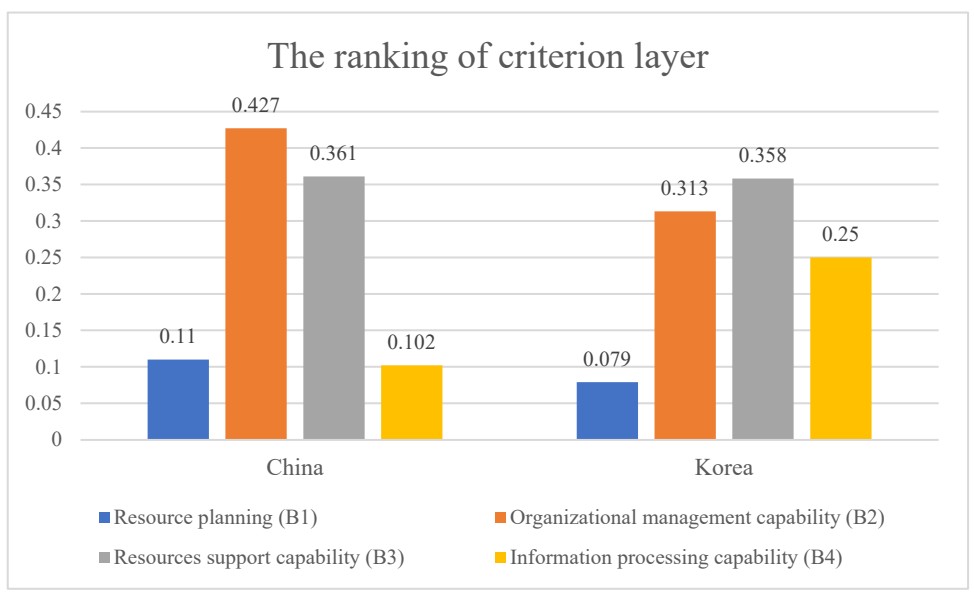

**Figure 3.** The criterion layer weight ranking of the operational capability of DEMRs.

Based on the judgment matrix, we can determine the relative importance of the third-level indicators compared to their respective second-level categories. For resource planning (B1), as illustrated in Figure 4, the importance rankings are identical for both China and Korea. Policy guidance (C1) holds the highest local importance, with weights of 0.425 for China and 0.312 for Korea, followed by demand assessment (C2) and funding budget (C4), with China's weights being 0.244 and 0.182, and Korea's at 0.279 and 0.222, respectively. Lastly, the resource operation plan (C3) has the lowest weights, at 0.149 for China and 0.187 for Korea.

For the organizational management capability (B2), as depicted in Figure 5, in China, the weight for fast response capability (C6) reached a significant 0.413, with command and dispatch capability (C5) also being crucial at 0.381. Additionally, social mobilization capability (C7) and communication capability (C8) were weighted at 0.142 and 0.063, respectively. Conversely, in Korea, fast response capability (C6) had a predominant local weight of 0.424, considerably outweighing communication capability (C8) at 0.244, command and dispatch capability (C5) at 0.218, and social mobilization capability (C7) at 0.114.

Regarding the resource support capability (B3), shown in Figure 6, for China, allocation capability (C12) with a weight of 0.451 and reserve capability (C9) with a weight of 0.325 were deemed most essential. Transportation capability (C10) and scheduling capability (C11) followed, with weights of 0.146 and 0.078, respectively. In Korea, allocation capability (C12) was also the most significant, with the highest local weight of 0.391, followed by reserve capability (C9) at 0.291 and scheduling capability (C11) at 0.217, with transportation capability (C10) receiving a lower priority.

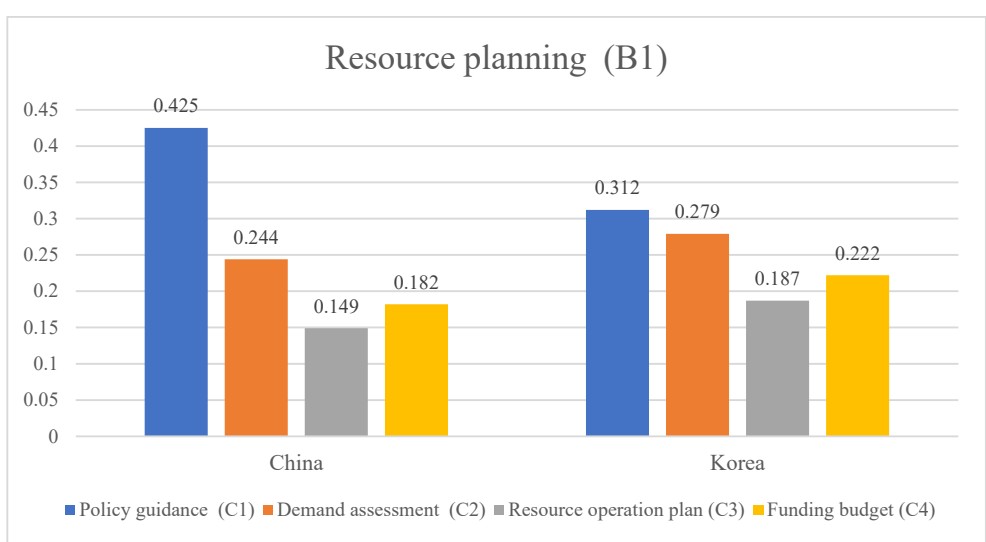

**Figure 4.** The local weight ranking of resource planning (B1) domain.

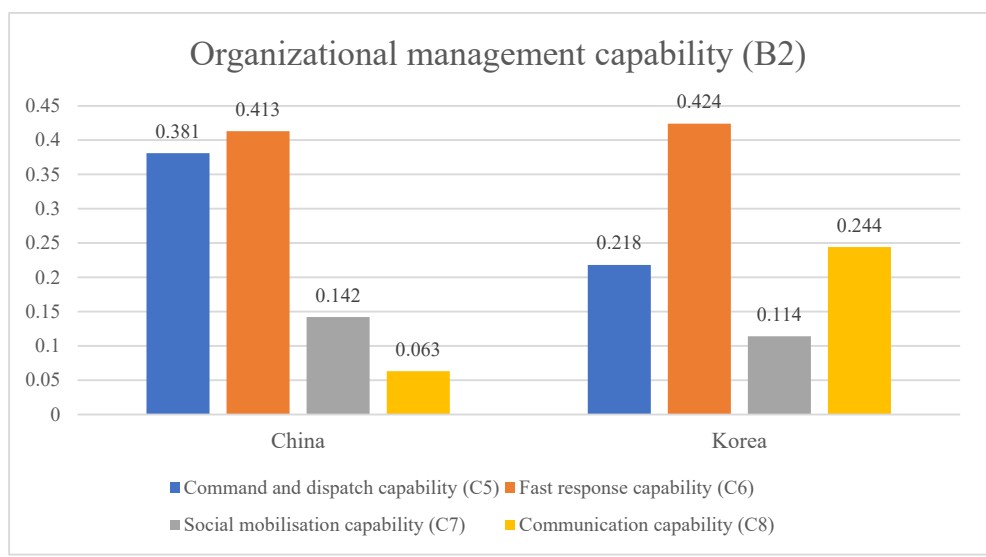

**Figure 5.** The local weight ranking of organizational management capability (B2) domain.

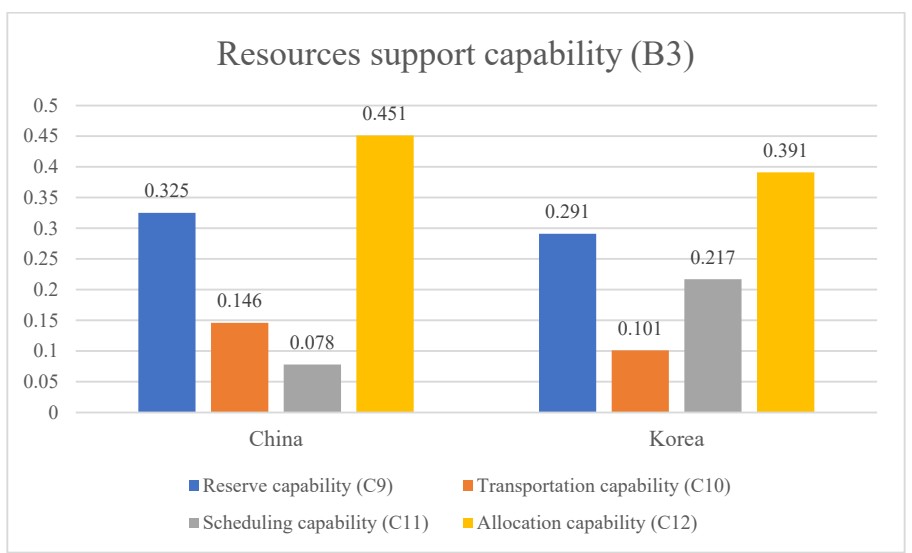

**Figure 6.** The local weight ranking of resources support capability (B3) domain.

In the context of information processing capability (B4), as illustrated in Figure 7, China and Korea show different prioritizations. In China, the monitoring and tracking of logistics (C16) plays a pivotal role in information processing capability, with a weight of 0.358. This is closely followed by timely information acquisition (C14) at 0.305, and early warning technology (C13) at 0.298, with only a slight difference between them. Information-sharing capability (C15) has the least weight.

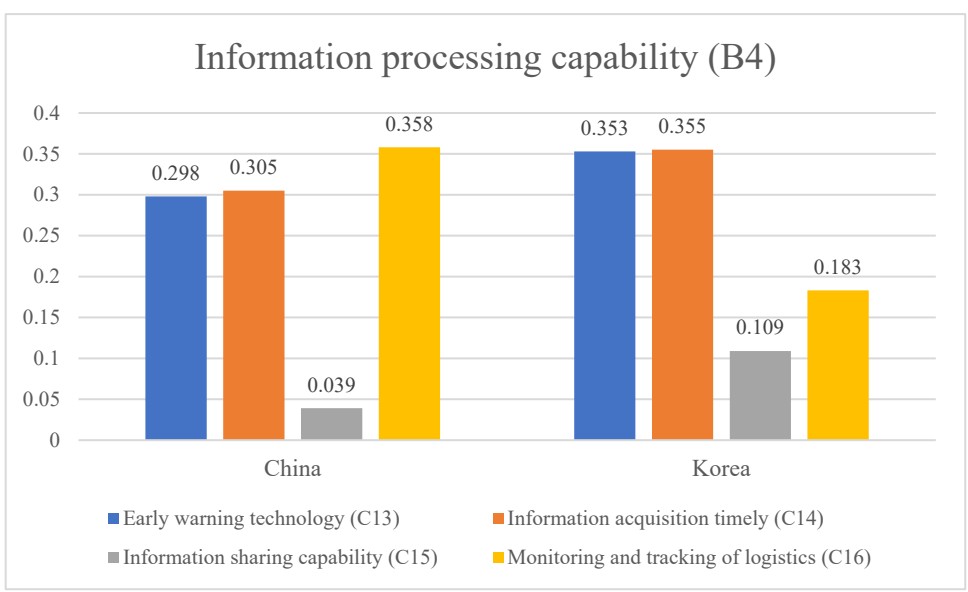

**Figure 7.** The local weight ranking of information processing capability (B4) domain.

In Korea, the weights for timely information acquisition (C14) and early warning technology (C13) are very close, at 0.355 and 0.353, respectively, indicating that it was challenging for the experts to distinguish between their relative importance due to their nearly equal significance. The monitoring and tracking of logistics (C16) follows with a weight of 0.183, and information-sharing capability (C15) has the lowest weight at 0.109, indicating that it is considered the least critical factor.

### 4.2. Global Weight Ranking Comparison

The global weight ranking provides insights into how various components within the scheme layer interact with both the overall goal and the scheme itself, often referred to as the composite or absolute weight ranking [2]. These global weights are calculated by multiplying the weight of each domain by the local weight of its indicators [53]. For example, the global weight for policy guidance (0.047) was derived by multiplying the weight of resource planning (0.11) by its local weight (0.425). Figure 8 illustrates the distribution of global weights for indicators within the scheme layer in relation to the target layer.

For China, as shown in Figure 8a, the top four indicators were fast response capability (0.176), command and dispatch capability (0.163), allocation capability (0.162), and reserve capability (0.117), all ranking within the top four. Following these, social mobilization capability (0.061) held the fifth position globally. The second-least prominent indicator was the resource operation plan (0.017), with information-sharing capability (0.004) being the least significant.

In Korea's case, as depicted in Figure 8b, allocation capability (0.140) emerged as the highest in global weighting, followed by fast response capability (0.133) and reserve capability (0.104). Notably, timely information acquisition (0.089) ranked fourth, and early warning technology (0.088) ranked fifth globally, contrasting with their ninth and tenth positions in China. This suggests that Korea places a greater emphasis on these two aspects regarding the operational capacity of DEMRs. The least significant indicators were funding

budget (0.018) and resource operation plan (0.015), highlighting different priorities between the two countries.

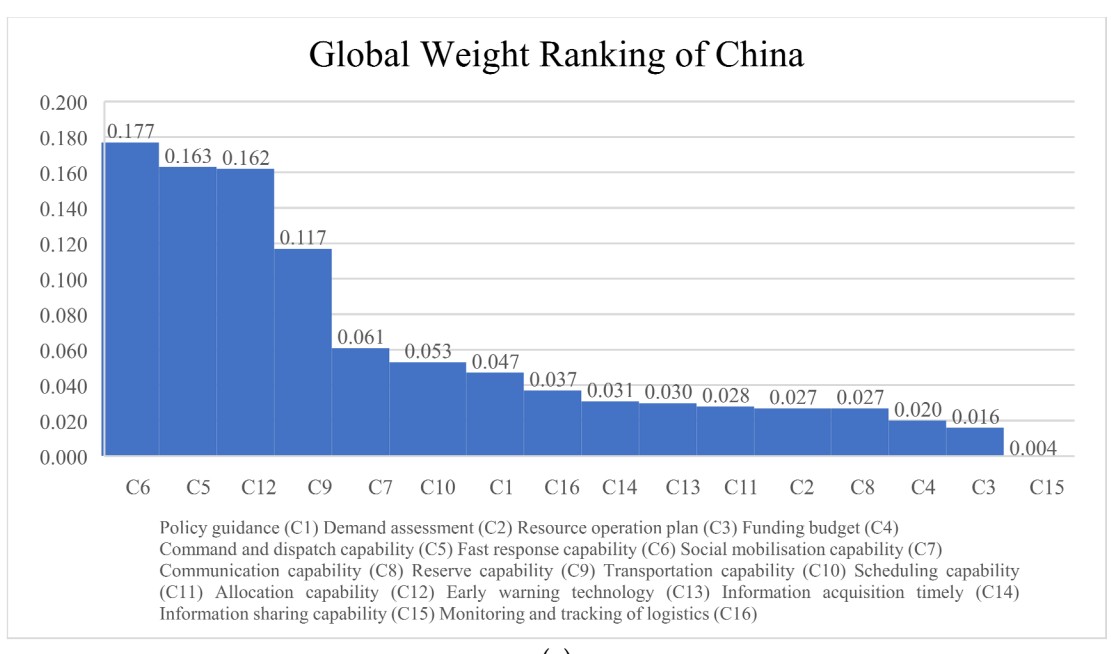

**(a)**

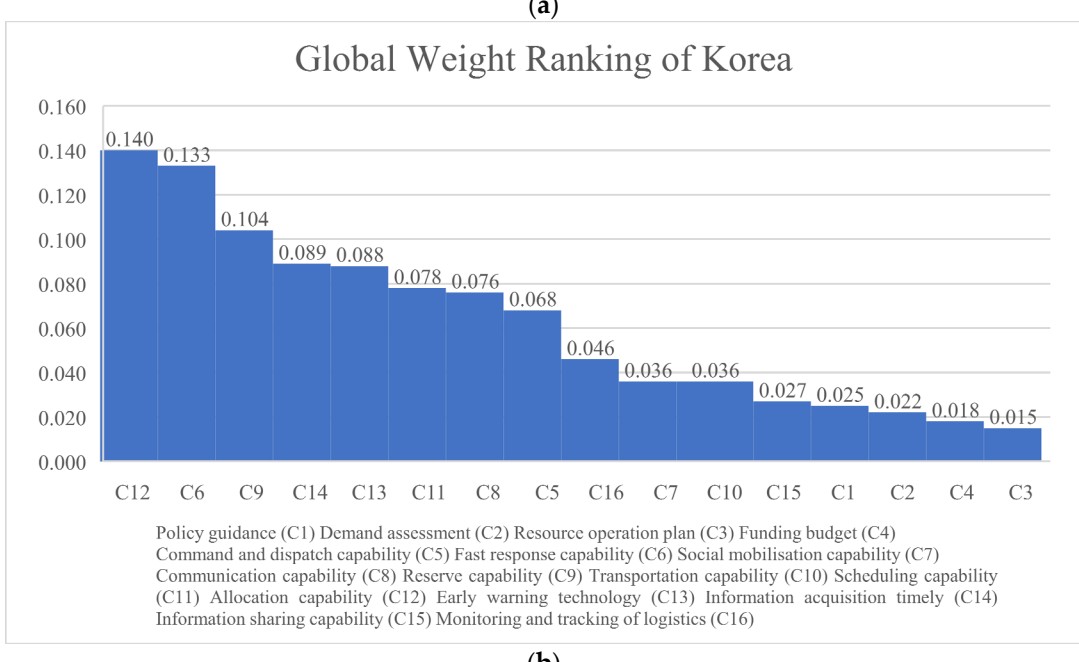

**(b)**

**Figure 8.** The global weight ranking of the operational capability of DEMRs. (**a**) The global weight ranking of China. (**b**) The global weight ranking of Korea.

### 4.3. Sensitivity Analysis

The aim of conducting a sensitivity analysis is to examine how the priorities of alternatives change with the priorities of criteria/sub-criteria [50]. Through sensitivity analysis, decision makers can grasp the influence of changes in attribute weights on decision results and the degree of that influence, helping decision makers make correct judgments.

In this study, we examine how the prioritization of domain weights varies according to changes in the weight of operational capability. As mentioned earlier, the findings showed that the weight of organizational management capability (B2) is ranked first in China. As shown in Figure 9a, the sensitivity analysis shows that fast response capability (C6) is the

most important indicator for the operational capability of DEMRs when the priority of organizational management capability (B2) is ≥0.42 (the rank reverse point, i.e., the dotted line); otherwise, allocation capability (C12) is the most important.

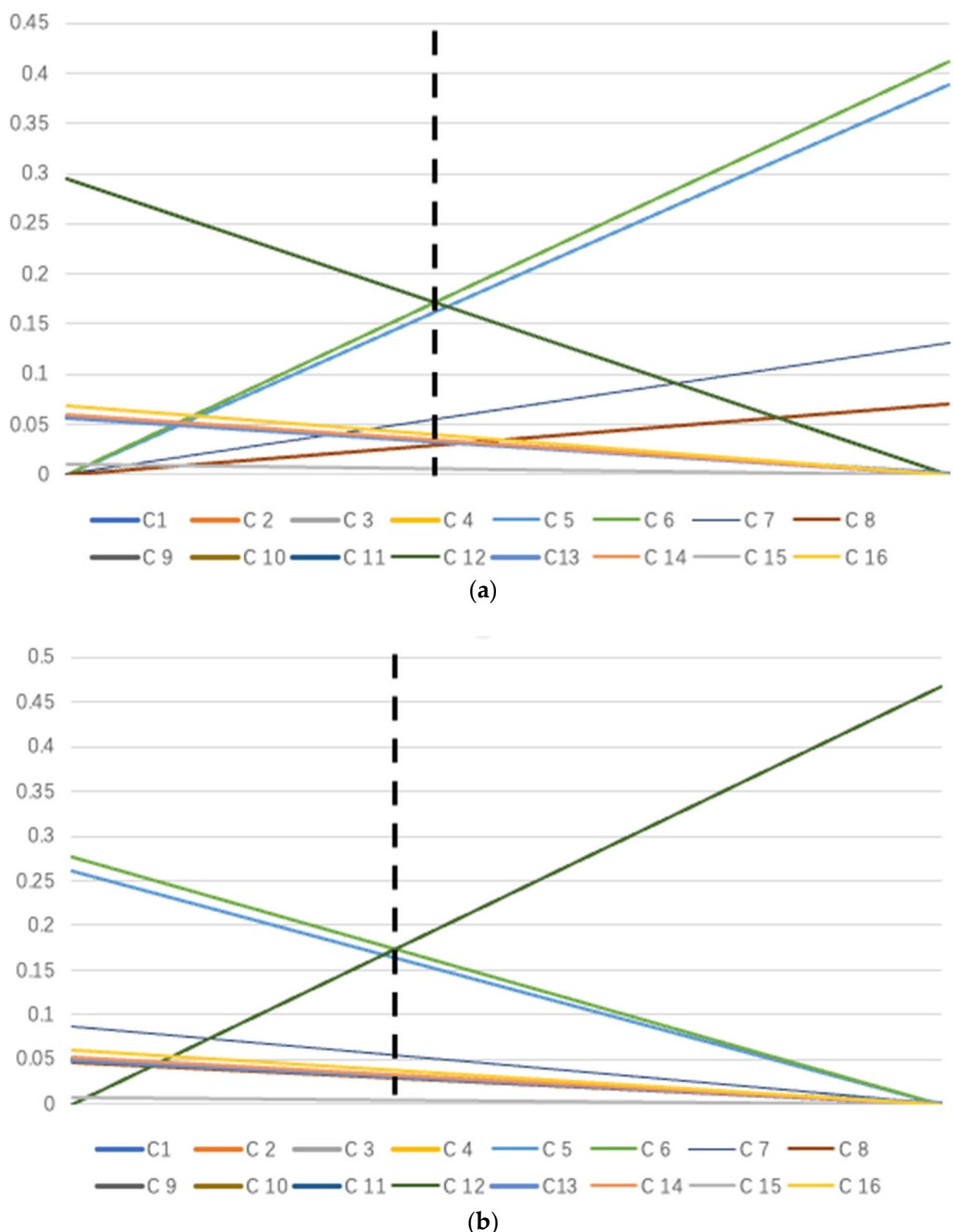

**Figure 9.** The results of sensitivity analysis. (**a**) The sensitivity analysis of organizational management capability (B2) in China. (**b**) The sensitivity analysis of resources support capability (B3) in Korea.

As mentioned earlier, the findings showed that the weight of resources support capability (B3) is ranked first in Korea. As shown in Figure 9b, the sensitivity analysis shows that fast response capability (C6) is the most important indicator for the operational capability of DEMRs, when the priority of resources support capability (B3) is ≤0.38 (the rank reverse point, i.e., the dotted line); otherwise, allocation capability (C12) is the most important.

## 5. Discussion

This study delineates the relative importance and priority of factors affecting the effectiveness of DEMRs in China and Korea, offering a detailed list of influential factors. The findings highlight several key insights: foremost, the analysis of domain weight and ranking indicates that organizational management capability holds the most significant impact on the operational capability of DEMRs in China, whereas resource support capability is prioritized in Korea. Additionally, information processing capability is ranked fourth in China but advances to third in Korea, suggesting a stronger emphasis on the role of information processing in Korea's effective management of DEMRs. This could be attributed to Korea's advanced use of information technology, including big data and the Internet of Things, propelled by its early adoption of the Fourth Industrial Revolution technologies compared to China. Despite some scholars highlighting information technology as a pivotal element for the operational capability of DEMRs [39,54], this study's findings do not entirely corroborate their views. While information technology has become crucial for disseminating disaster-related information since the advent of the Fourth Industrial Revolution, its implementation has sometimes failed to foster adequate collaboration between governments and agencies. This lack of synergy has occasionally hindered resources from being delivered in a timely manner to disaster-stricken areas [55].

At the same time, it appears that both China and Korea have underestimated the importance of resource planning in the functionality of their DEMRs. According to Hu et al. [56], following the issuance of the Measures for Administration of Emergency Management Plans, the Chinese government has made significant strides in enhancing plan formulation and implementation, improving disaster preparedness via drills and training, and strengthening local emergency management capabilities. However, these plans still face challenges, such as flaws and a lack of standardization. Bae et al. [57] have also pointed out the insufficient comprehensive emergency planning in Korea, with disaster management resources often being allocated the lowest priority in existing emergency strategies, especially when compared to plans focusing on economic development.

Furthermore, when examining the ranking of local weights across each evaluation domain, policy guidance emerged as the paramount factor in both China and Korea in the realm of resource planning. This aligns with findings that suggest that effective policies can better prepare communities to respond to disasters [58], thereby enabling stakeholders to be more proactive and prepared. Given the pivotal role of policy guidance in the operational dynamics of DEMRs and overall disaster management, it is imperative for public organizations and policymakers to consider revisions to public policy and practices. Such changes should aim to enhance the capacity to effectively manage future disasters, drawing on the lessons learned from past experiences [59].

Concerning organizational management capability, both countries prioritize fast response as crucial. This underscores the importance of swiftly addressing the needs for emergency relief in disaster-stricken areas, highlighting it as a vital aspect of effective operation of DEMRs. During the evaluation of resource operations, the efficiency of fast response should be considered a key metric, especially in the context of urgently allocating resources to essential systems and the limited recovery time [60].

In the realm of resource support capability, both countries recognize the resource allocation capability and reserve capability as critical factors influencing the operational capability efficiency of DEMRs, with the allocation capability being deemed more crucial than reserve capability. As Rodríguez-Espíndola [24] has shown, the lack of rational resource allocation can result in poor emergency responses, even when resources are plentiful.

Significant differences were observed between the two countries in terms of their information processing capabilities. In China, the most impact was seen in the monitoring and tracking of logistics, whereas in Korea, this was less pronounced. Meanwhile, the importance of timely early warning systems and access to information was almost equally recognized by both. The stark contrast in land size between China and Korea, with China's extensive territory and the logistical challenges of transporting resources over long

distances, necessitates the use of mobile information gathering devices like GPS and GPRS for resource positioning and real-time tracking [61].

Thirdly, the comparison of global weight rankings indicates that fast response capability, resource allocation capability, and reserve capability were highly valued in both countries, highlighting that these three factors had a more significant impact on the operational capability of DEMRs. Furthermore, the capability for command and dispatch was rated more prominently in China than in Korea, securing the second position in the global ranking. This reflects China's disaster management approach [62], which is characterized by unified leadership and a tiered response system. In this framework, the government is responsible for policy formulation, decision making, and the coordination of disaster management efforts. In contrast, Korea places a greater emphasis on the capacity of information access and early warning technologies, ranking these significantly higher than China. This difference underscores Korea's focus on the importance of information processing in resource management. With advancements in big data and AI technology, the management of information resources has become increasingly vital. To support emergency management decisions and efficiently manage resources, it is essential to swiftly and effectively collect, integrate, analyze, and utilize various information processing capabilities [63]. China could benefit from adopting Korea's approach, enhancing its collection of emergency information, maintaining alertness to significant disaster signals, improving its information analysis capabilities, and ensuring the timely dissemination of early warnings.

In the global weight ranking comparison between the two countries, it was observed that funding budgets and resource operation plans were deemed less critical. These elements are integral to emergency planning, and as Hu et al. [56] have highlighted, emergency planning faces obstacles such as limited funding and underdeveloped strategies. Consequently, these factors contribute less significantly to emergency response efforts than others. Furthermore, in China, the capacity for information sharing was ranked as the least important, even more so than in Korea. This discrepancy arises because various departments, local governments, and agencies often operate their own information databases and systems without a centralized platform for sharing information. This fragmentation can result in the undervaluation of information sharing capabilities.

## 6. Conclusions

This study constructed an evaluation index system for assessing the relative importance and priority of various factors and indicators impacting the operational capability of DEMRs in China and Korea, facilitating a detailed comparative analysis. The results revealed that organizational management capacity (B2) and resource support capacity (B3) are the most heavily weighted domains in operational capacity for China and Korea, with weight values of 0.427 and 0.358, respectively. Meanwhile, both countries identify fast response capability, resource allocation capability, and reserve capability as key impact indicators. However, both countries neglect the importance of funding budget and information sharing capacity, with information sharing capacity accounting for only 0.004 of the global weight in China and 0.015 of the funding budgets in Korea. A notable distinction is Korea's higher prioritization of information processing capabilities, with early warning systems and timely access to information being ranked significantly higher than in China. In contrast, China places more emphasis on the command and dispatch capability in the operational capability of DEMRs, ranking second with a weight of 0.163.

This study not only reveals the indicators affecting the operational capability of DEMRs, but also points out the shortcomings of existing operational procedures. In terms of the ranking of the weights of indicators, the lack of attention to resource planning and information processing capability in the operational capability of DEMRs in China, and the same lack of attention to resource planning on the part of Korea, may lead to a shortage of resources in disaster relief activities, and an imbalance between supply and demand. The results of this study can help to identify weaknesses in the operation of future DEMRs and facilitate evidence-based decision making and policy formulation. Effective DEMR

operation not only mitigates immediate risks and minimizes losses but also promotes long-term resilience and sustainability by preserving infrastructure, ecosystems, and livelihoods.

The comparative analysis of China and Korea conducted in this study provides valuable insights into regional differences in disaster management priorities and practices, thereby providing targeted strategies for enhancing sustainability at the national and international levels. For managers, leveraging the operational strengths observed in other countries can significantly enhance the efficiency of their own DEMRs, and build more resilient and sustainable societies.

Future studies should delve deeper into the variances in disaster and emergency resources across different nations, aiming to develop a more comprehensive and scientific evaluation framework. Such efforts will enable a more precise assessment of the operational capability of DEMRs worldwide, contributing to the enhancement in global disaster and emergency resource management levels.

**Author Contributions:** Conceptualization, K.Z. and J.E.L.; methodology, K.Z.; writing—original draft preparation, K.Z.; writing—review and editing, K.Z. and J.E.L.; visualization, K.Z.; supervision, J.E.L.; funding acquisition, J.E.L. All authors have read and agreed to the published version of the manuscript.

**Funding:** This work was supported by the Ministry of Education of the Republic of Korea and the National Research Foundation of Korea (NRF-2023S1A5C2A02095270).

**Institutional Review Board Statement:** Not applicable.

**Informed Consent Statement:** Informed consent was obtained from all subjects involved in the study.

**Data Availability Statement:** Data are contained within the article.

**Conflicts of Interest:** The authors declare no conflicts of interest.

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
