# Peer review of "Assessing the Operational Capability of Disaster and Emergency Management Resources: Using Analytic Hierarchy Process"

_sustainability, doi:10.3390/su16103933_

Round 1

Reviewer 1 Report

Comments and Suggestions for Authors

Dear Authors

 I reviewed your manuscript, I appreciate your efforts to complete the research work, I have following comment to improve the article.

1. Introduction is well written.

2.      The literature review is short, it requires to include some more recent work.

3.      The methodology (AHP) specifically adopted in the research should be more clearly defined.  There is no discussion about the scale used here. The following  related reference will help you.

https://www.emerald.com/insight/content/doi/10.1108/IJM-04-2020-0164/full/html

http://dx.doi.org/10.21511/ppm.22(1).2024.37

Reviewer 2 Report

Comments and Suggestions for Authors

The authors propose the use of AHP to establish importance figures among different operational criteria for measuring capability in disaster and emergency management, which is an intriguing idea.

The author also proposes a comparative analysis between Chinese and Korean perceptions regarding the relative importance among the criteria. However, it is not clear why this specific comparison is important, especially considering other countries with extensive experience in disaster management.

The author asserts that there is an innovation "by incorporating coordinated strategies for resource planning (...)." However, it is not explicitly explained how this was accomplished. It should be clearly addressed, perhaps in the Materials and Methods section.

In Section 3, the authors explain the importance of considering expert opinions for pairwise comparisons in the AHP procedure. They mention two groups of experts: the first composed of ten PhD researchers and the second composed of 22 experts in the field of emergency and disaster management. It would be beneficial to specify the characteristics of these experts. How were the PhD researchers selected? What background qualifications were necessary to be part of this expert group? The same clarification should be provided for the 22 country experts (11 from China and 11 from Korea).

In traditional AHP procedures, comparison matrices are typically singular (based on a “single opinion”). Considering the multitude of experts consulted in this study, how did the authors combined the various opinions to obtain a singular matrix for each pairwise comparison? There are numerous research papers proposing different approaches for group decision-making using the AHP methodology. On what basis did the authors develop their proposition?

In the conclusion section, the authors suggest that the proposed approach "pinpoints shortcomings in existing operational procedures." It would be beneficial if these shortcomings were presented in a tabular format.

Reviewer 3 Report

Comments and Suggestions for Authors

The paper number 2956239 is interesting and within the scope of the Journal. The title reflects the content of the manuscript Assessing the Operational Capability of Disaster and Emergency Management Resources: Using Analytic Hierarchy Process (AHP).

I suggest the authors summarizing the first and second chapter into only one: the introduction. The state of the art should appear before the lines 87-101 that are the end of the Introduction. Moreover, they should specify why the experts are appropriate (line 224). Which is their work? Which is their expertise? Are they male or female? How long is their experience?

Consider deepening the literature review with references on existing methodologies for evaluating the minimum road network with connectivity and availability to guaranty main urban facilities access in seismic events (pre-emergency planning). (Road) lifelines are pivotal for such issue: 10.17226/22338 and 10.3390/su132011151

The authors can use programs as Expertchoice or Superdecisions to make a sensitivity analysis of the results.

The authors should use only one English language: British (with "s") or American (with "z").

Avoid the acronym DEMR in the keywords

Some quantitative results should be discussed in the Conclusions.

Comments on the Quality of English Language

British or American English?

Round 2

Reviewer 2 Report

Comments and Suggestions for Authors

Once again, I appreciate the opportunity to contribute to the review of the referred paper. I also thanks the authors for considering the proposed adjustments made in my the initial review. I believe the paper is now suitable for publication in the Sustainability journal. Best regards!

Reviewer 3 Report

Comments and Suggestions for Authors

the paper can be accepted